# Comprehensive multi-omics analysis reveals prognostic, immune, and therapeutic signatures of TNFAIP family genes in breast cancer

Tahereh Barati[1☝], Madiheh Mazaheri Moghaddam[2☝], Fatemeh Mokhles[3], Zohreh Mirzaei[1], Amir Ebrahimi[1], Golsa Nayeb Ghanbar Hosseini[4], Najaf Allahyari Fard[5]*

1 Department of Medical Genetics, Faculty of Medicine, Tabriz University of Medical Sciences, Tabriz, Iran, 2 Department of Genetics and Molecular Medicine, School of Medicine, Zanjan University of Medical Sciences, Zanjan, Iran, 3 Department of Molecular Medicine, Institute of Medical Biotechnology, National Institute of Genetic Engineering and Biotechnology (NIGEB), Tehran, Iran, 4 Institute of Biology, Leiden University, Leiden, Netherland, 5 Systems Biotechnology Department, National Institute for Genetic Engineering and Biotechnology, Tehran, Iran

☝ These authors contributed equally to this work.
* allahyar@nigeb.ac.ir

## Abstract

### Background

Tumor necrosis factor-alpha-induced proteins (TNFAIPs) are key regulators of inflammation, apoptosis, and immune signaling, yet their integrated roles in breast cancer (BC) remain poorly characterized. While individual TNFAIP members have been studied in other cancers, a comprehensive multi-omics characterization of this gene family in BC is still needed.

### Methods

We performed an integrative bioinformatics analysis using public datasets (UALCAN, TIMER, bc-GenExMiner, cBioPortal, STRING, GeneMANIA, Enrichr, MethSurv, GDSC, CTRP, and HPA) to evaluate TNFAIP family members in BC. Expression, genomic alterations, methylation patterns, immune infiltration, and drug sensitivity were analyzed across clinical and molecular subtypes. Where applicable, multiple hypothesis testing was controlled using the Benjamini–Hochberg false discovery rate (FDR) method.

### Results

Among TNFAIPs, TNFAIP6 and EFNA1 were significantly upregulated in BC, while TNFAIP1, TNFAIP2, TNFAIP3, PTX3, TNFAIP8, and STEAP4 were downregulated. Elevated TNFAIP2, TNFAIP3, and TNFAIP8 expression correlated with improved overall survival (OS). Multi-database integration revealed that TNFAIP3 expression

**Data availability statement:** All relevant data are within the manuscript and its Supporting Information files.

**Funding:** The author(s) received no specific funding for this work.

**Competing interests:** The authors declare that there are no competing interests.

was strongly correlated with infiltration of CD4$^+$T cells, dendritic cells, and neutrophils. Functional enrichment highlighted the NF-κB, PI3K-Akt, and TNF signaling pathways as key regulatory axes. Drug-sensitivity analyses indicated subtype-dependent responses linked to TNFAIP dysregulation.

## Conclusion

This study provides a comprehensive multi-omics characterization of TNFAIP family genes in BC, addressing the role of inflammatory signaling in tumor progression and identifying potential biomarkers and therapeutic targets. These findings enhance the understanding of TNFAIP-mediated molecular networks and offer a resource for translational and experimental research in BC.

---

## 1. Introduction

Breast cancer (BC) is currently the most frequently diagnosed malignant cancer in the world, accounting for 11.7% of all cases. It also ranks as the fifth leading cause of cancer-related death (~6.9% of all cases) [1,2]. BC presents a genetically diverse type of tumor, with a range of morphological characteristics and is classified into five types: Luminal A and B, Triple-negative/Basal-like, human epidermal growth factor 2 (HER2)-positive, and normal-like that differ in their molecular signatures, immune landscapes, and therapeutic responses [3,4]. Therefore, understanding gene family behavior is critical for identifying subtype-specific biomarkers and treatment strategies. Although there has been improvement in the diagnosis and treatment of BC in recent decades, the most significant challenge in clinical treatment continues to be the incurable nature of metastasis and recurrence [5].

Tumor Necrosis Factor (TNF), a well-known inflammatory factor, exists in soluble and transmembrane forms that signal through TNFR1 and TNFR2 to regulate diverse cellular processes including inflammation, apoptosis, and immune modulation [6]. Upon TNF stimulation, a family of tumor necrosis factor alpha-induced proteins (TNFAIPs) is expressed, acting downstream to mediate or regulate TNF-driven responses [7]. The TNFAIP family comprises eight members, including TNFAIP1, TNFAIP2, TNFAIP3 (also known as A20), TNFAIP4 (hereafter referred to as EFNA1), TNFAIP5 (hereafter referred to as PTX3), TNFAIP6, TNFAIP8, and TNFAIP9 (hereafter referred to as STEAP4) [8]. According to earlier research, TNFAIPs frequently play important roles in cell differentiation, apoptosis, signal transduction, inflammation, immune response, and other biological functions. Additionally, they are crucial in the pathogenesis of illnesses, particularly cancerous tumors [9–11].

Previous investigations have described the expression patterns of various TNFAIPs in BC [12–14]. However, the global molecular landscape and clinical relevance of TNFAIP family members in BC remain incompletely characterized. Recent studies suggest that members of the TNFAIP family, particularly TNFAIP3, play critical roles in regulating NF-κB signaling, immune responses, and tumor progression. For instance, TNF-α promotes BC cell growth by activating a positive feedback loop

involving TNFR1, NF-κB, STAT3, and the oncoprotein HBXIP [15]. Moreover, TNFAIP3 not only suppresses inflammatory signaling but also enhances angiogenesis via interaction with FGFR1 and upregulation of VEGFA expression [14]. Also, TNFAIP2 promotes triple-negative breast cancer (TNBC) angiogenesis through the Rac1-ERK-AP1-HIF1α signaling axis under hypoxia, but the complete regulatory network and upstream factors remain to be mapped [16].

Despite emerging evidence highlighting the role of certain TNFAIP members like TNFAIP3 in NFκB regulation [14,15] or TNFAIP2 in cell migration and formation of tunneling nanotubes [12,17], the overall expression landscape, methylation status, subtype-specific patterns, prognostic significance, immune associations, and drug sensitivity profiles of the full TNFAIP family in BC remain largely unexplored [18,19]. Therefore, our study provides a comprehensive multi-dimensional analysis of the TNFAIP family in BC within an integrative framework, aiming to identify potential prognostic and therapeutic signatures across clinical and molecular contexts.

## 2. Materials and methods

To comprehensively investigate the expression patterns, prognostic significance, and functional implications of the TNFAIP gene family in BC, we employed a multi-dimensional bioinformatics approach utilizing publicly available genomic, transcriptomic, and epigenomic databases. This integrated strategy allowed us to analyze gene expression profiles, assess clinical correlations, evaluate immune microenvironment interactions, identify genomic alterations, and predict therapeutic vulnerabilities. The selection of these specific bioinformatics tools was based on their established reliability, extensive use in cancer research, and ability to provide complementary insights through specialized analytical capabilities.

### 2.1. Gene expression profiling using UALCAN

To assess the transcriptional expression of TNFAIP family members in BC and corresponding normal tissues, we utilized the UALCAN platform (http://ualcan.path.uab.edu/) [20]. This publicly accessible resource integrates high-throughput OMICS data from The Cancer Genome Atlas (TCGA), enabling robust differential gene expression analysis across multiple cancer types [21]. We employed the "TCGA gene analysis" module to compare mRNA expression levels of TNFAIP genes between tumor and normal breast samples. Additionally, we evaluated associations between TNFAIP expression and clinical parameters, including tumor stage and molecular subtypes, to elucidate their potential clinical relevance. Statistical significance was defined based on the criteria implemented within the UALCAN platform ($p < 0.05$). As this analysis focused on a predefined TNFAIP gene family, results were interpreted within a hypothesis-driven framework [22].

### 2.2. Immune infiltration analysis via TIMER

The Tumor Immune Estimation Resource (TIMER; https://cistrome.shinyapps.io/timer/) was employed to investigate the relationship between TNFAIP expression and immune cell infiltration within the tumor microenvironment. TIMER applies a deconvolution algorithm to estimate the abundance of six immune cell types (B cells, CD4+T cells, CD8+T cells, neutrophils, macrophages, and dendritic cells) based on TCGA gene expression profiles [23]. Using the "DiffExp" module, we compared TNFAIP expression between tumor and adjacent normal tissues. Correlation analyses between TNFAIP expression and immune cell infiltration levels were conducted using the TIMER platform. Statistical significance was interpreted according to the platform's implemented statistical framework ($p < 0.05$), and correlation patterns were evaluated in an exploratory context [24].

### 2.3. Clinicopathological correlation using bc-GenExMiner v5.0

We utilized bc-GenExMiner v5.0 (http://bcgenex.ico.unicancer.fr/BC-GEM/GEM-Accueil.php/) to examine the relationship between TNFAIP expression and key clinicopathological features, including Scarff-Bloom-Richardson (SBR) grade and Prediction Analysis of Microarray 50 (PAM50)-based molecular subtypes. This platform aggregates curated transcriptomic

and clinical data from multiple BC studies, facilitating integrated bioinformatics analyses. Statistical comparisons were performed using Welch's t-test and the Dunnett-Tukey-Kramer test. Statistical significance was defined according to the analytical framework implemented within the bc-GenExMiner platform (p < 0.05) [25].

### 2.4. Survival Analysis with Kaplan-Meier Plotter

Prognostic implications of TNFAIP expression were evaluated using the Kaplan–Meier Plotter (http://kmplot.com/analysis/) a database linking gene expression data to clinical outcomes across 21 cancer types. Patients were stratified into high- and low-expression groups based on median mRNA levels. Overall survival (OS) and relapse-free survival (RFS) were compared using hazard ratios (HR) with 95% confidence intervals (CI) and log-rank tests. To account for multiple comparisons across TNFAIP family members, FDR correction using the Benjamini–Hochberg method was applied, and FDR < 0.05 was considered statistically significant [26].

### 2.5. Genomic alteration profiling via cBioPortal

The cBioPortal platform (http://cbioportal.org/) was used to analyze genomic alterations, including mutations, copy number alterations (CNAs), and mRNA expression Z-scores within the TNFAIP family in a TCGA breast invasive carcinoma cohort (n = 1,108). We also extracted the top 50 frequently altered genes co-occurring with TNFAIP members to identify potential functional partners. Statistical significance was interpreted based on the criteria implemented within the cBioPortal platform (p < 0.05) [27].

### 2.6. Functional interaction networks with GeneMANIA and STRING

GeneMANIA (http://www.genemania.org/) was used to construct a gene-gene interaction network for TNFAIP members, identifying functionally related genes based on co-expression, physical interactions, and pathway sharing [28]. Additionally, protein-protein interaction (PPI) networks were generated using STRING (https://string-db.org/) with a confidence score threshold > 0.4. Resulting networks were visualized and analyzed using Cytoscape v3.10.1 [29,30].

### 2.7. Enrichment analysis using Enrichr

Functional enrichment analysis for Gene Ontology (GO) terms, Kyoto Encyclopedia of Genes and Genomes (KEGG) pathways, transcription factors, and miRNA interactions was performed using Enrichr (https://maayanlab.cloud/Enrichr/). For GO and KEGG enrichment, terms were considered significant based on p < 0.05. For TF and miRNA enrichment, significance was determined using FDR-adjusted p-values (FDR < 0.05) to control for multiple testing. results were visualized using the ggplot2 package in R [31].

### 2.8. DNA methylation analysis with MethSurv

MethSurv (https://biit.cs.ut.ee/methsurv/) was used to evaluate the prognostic value of CpG methylation sites within TNFAIP genes. Methylation levels and their association with patient survival were analyzed using the LR test. To control for multiple testing, FDR-adjusted p-values were calculated, and significance was defined as FDR < 0.05 [32].

### 2.9. Drug sensitivity assessment via GSCALite

Drug sensitivity profiles for TNFAIP family members were analyzed using GSCALite (https://github.com/chunjie-sam-liu/GSCALite/), which integrates data from the Genomics of Drug Sensitivity in Cancer (GDSC) and Cancer Therapeutics Response Portal (CTRP). Correlations between gene expression and drug response (IC$_{50}$ values) were assessed to identify potential therapeutic implications [33].

## 2.10. Qualitative IHC analysis using the human protein atlas (HPA)

Qualitative immunohistochemical (IHC) analysis was performed using publicly available images from the HPA database (https://www.proteinatlas.org/). IHC staining patterns for TNFAIP family were reviewed in tumor tissues and matched normal counterparts across available cancer types. Representative high-resolution images were selected to illustrate protein localization and relative staining intensity. Staining was evaluated qualitatively based on the annotation provided by the HPA (negative, weak, moderate, or strong) and by visual comparison of cytoplasmic, nuclear, and membranous patterns. Only antibody versions validated in the HPA for immunohistochemistry were included.

## 2.11. Statistical considerations

All statistical analyses were conducted using default parameters within each tool. For comparisons involving multiple tests, FDR correction based on the Benjamini–Hochberg method was applied where applicable. For analyses where FDR was not available, raw p-values were reported and interpreted with caution. Significance was defined as FDR < 0.05 when multiple testing correction was applied, and nominal $p < 0.05$ otherwise.

## 2.12. Ethical approval

This study used publicly available datasets and did not involve human participants or animal experiments; therefore, no ethical approval was required.

## 3. Results

### 3.1. Expression analysis of the TNFAIPs in patients with BC

By using the TIMER and UALCAN datasets, we investigated the expression of the TNFAIP family in patients with BC. According to TIMER database results, the expression of EFNA1 and TNFAIP6 in primary tumors was higher than in normal tissues, while TNFAIP1, TNFAIP2, TNFAIP3, PTX3, TNFAIP8, and STEAP4 were significantly lower in tumor samples (all p < 0.05) (Fig 1A). We also evaluated the expression of the TNFAIP family in normal breast tissues and BC tissues using the UALCAN database (Fig 1B). The findings were consistent with those found in the TIMER database (all p < 0.05).

### 3.2. Clinicopathological characteristics of BC patients and the expression of the TNFAIPs

Using the UALCAN and Bc-GenExMiner databases, we next evaluated the association between the expression of the TNFAIP family and the clinicopathological characteristics of individuals with BC. Table 1 illustrates the upregulated expression of TNFAIP3, PTX3, and TNFAIP6 in the ≤ 51 years old group relative to the > 51 years old group (p < 0.05). In patients with BC, the mRNA levels of STEAP1 were found to be higher in positive lymph nodes as compared to negative lymph nodes, while the expression levels of TNFAIP2 and PTX3 were low (p < 0.05). The mRNA levels of TNFAIP2, TNFAIP3, EFNA1, PTX3, and TNFAIP8 were found to be higher in ER-negative BC. On the other hand, the mRNA level of TNFAIP6 was higher in ER-positive BC (p < 0.05). Moreover, in PR-negative BC patients, expression levels of TNFAIP2, TNFAIP3, PTX3, and TNFAIP8 were elevated, and in PR-positive BC, expression levels of TNFAIP6 and STEAP4 were increased. TNBC was found to have a significant correlation with increased levels of TNFAIP2, TNFAIP3, EFNA1, PTX3, TNFAIP6, TNFAIP8, and decreased levels of TNFAIP1 and STEAP4 (p < 0.05). The expression levels of TNFAIP1, TNFAIP3, PTX3, and TNFAIP8 significantly decreased as the tumor stage increased. Meanwhile, the expression of EFNA1 increased as the tumor progressed to a more advanced stage (Fig 2A). Concerning the molecular PAM50 subtypes of BC, TNFAIP2, TNFAIP3, and PTX3 expression was significantly higher in patients with TNBC compared to the other subtypes (basal-like, luminal A/B, and normal breast-like (p < 0.05)). The expression of TNFAIP1 was notably higher in HER2-positive (p < 0.05) and the expression of EFNA1 and TNFAIP6 was higher in the luminal A subtype in comparison to other subtypes (Fig 2B). As seen in Fig 2C, there

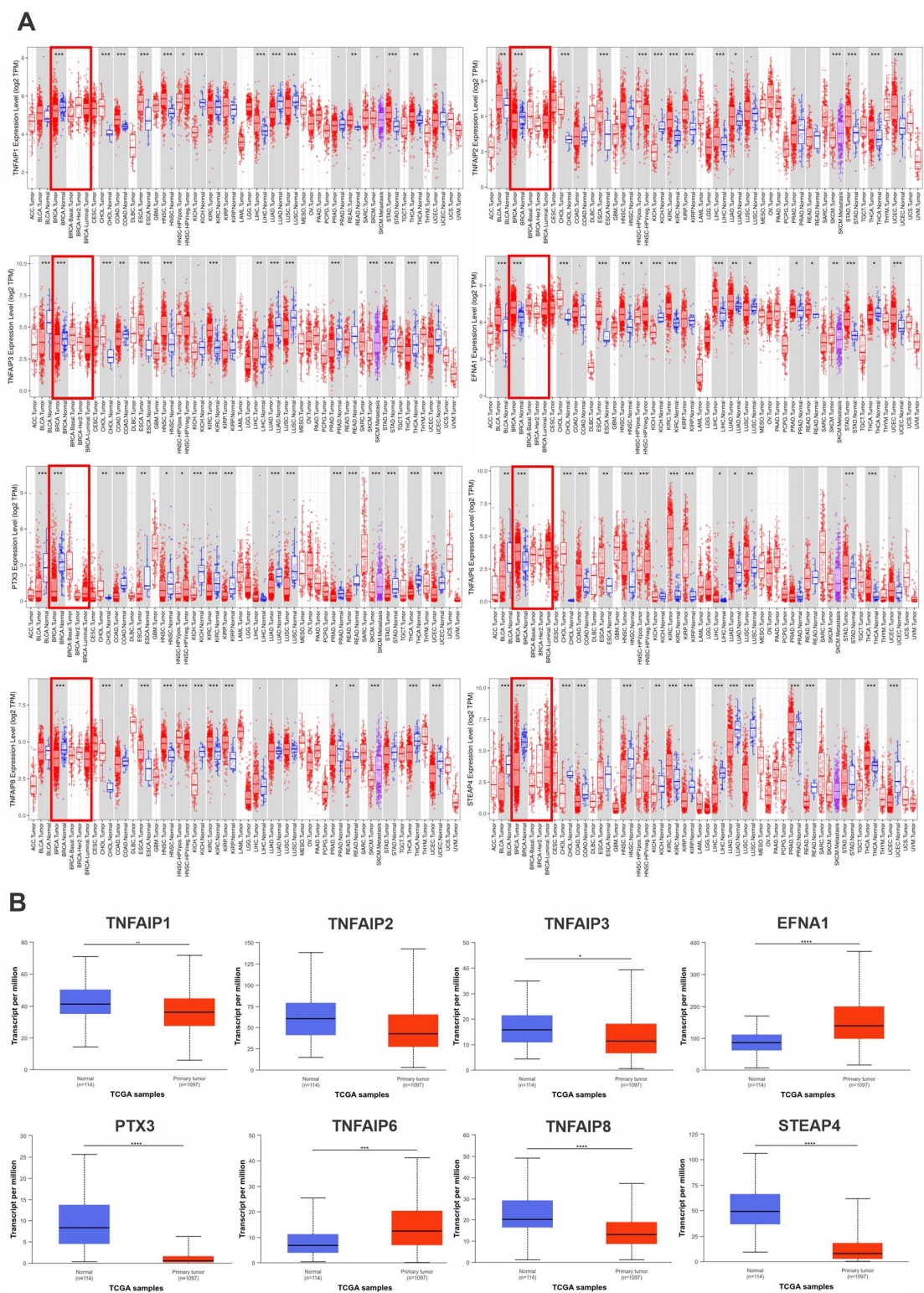

**Fig 1. mRNA expression profiles of TNFAIP family members in BC tumor and normal tissues (A: TIMER Database; B: UALCAN Database).**
**A**: The y-axis indicates log2 TPM expression levels, while the x-axis lists TCGA samples across tumor and normal tissues. Red lines represent mRNA

expression in tumor samples, and blue lines represent expression in normal samples, with blue regression lines showing the trend. **B**: The y-axis represents transcripts per million (TPM) expression levels, while the x-axis categorizes samples into Normal and Primary Tumor groups. Red boxes indicate expression in tumor samples, and blue boxes indicate expression in normal samples, with boxes showing the median and interquartile range. *P < 0.05, **P < 0.01, ***P < 0.001, ****P < 0.0001.

**Table 1.  The relationship between the mRNA levels of TNFAIP family genes and the clinicopathological features of BC patients. (bc-GenExMiner v5.0).**

| Gene | Criteria | Age ≤51 >51 | Nodal status (-) (+) | ER(IHC) (-) (+) | PR(IHC) (-) (+) | HER2 (IHC) (-) (+) | TNBC Not TNBC | Basal-like Not Basal-like |
|---|---|---|---|---|---|---|---|---|
| TNFAIP1 | NO<br>mRNA<br>P.value | 267 476<br>- -<br>0.5214 | 332 358<br>- -<br>0.2450 | 187 530<br>- -<br>0.1194 | 243 470<br>- -<br>0.2677 | 396 109<br>- ↑<br>**<0.0001*** | 578 87<br>↑ -<br>**<0.0001*** | 605 136<br>↑ -<br>**<0.0001*** |
| TNFAIP2 | NO<br>mRNA<br>P.value | 267 476<br>- -<br>o.9402 | 332 358<br>- ↓<br>**0.0180*** | 187 530<br>↑ -<br>**0.0046*** | 243 470<br>↑ -<br>**0.0053*** | 396 109<br>↑ -<br>**0.0139*** | 578 87<br>- ↑<br>**0.0005*** | 605 136<br>- ↑<br>**<0.0001*** |
| TNFAIP3 | NO<br>mRNA<br>P.value | 267 476<br>↑ -<br>**0.0030*** | 332 358<br>- -<br>0.2915 | 187 530<br>↑ -<br>**<0.0001*** | 243 470<br>↑ -<br>**0.0002*** | 396 109<br>↑ -<br>**0.0290*** | 578 87<br>- ↑<br>**<0.0001*** | 605 136<br>- ↑<br>**<0.0001*** |
| EFNA1 | NO<br>mRNA<br>P.value | 267 476<br>- -<br>0.4675 | 332 358<br>- -<br>0.4785 | 187 530<br>↑ -<br>**0.0424*** | 243 470<br>- -<br>0.4191 | 396 109<br>- -<br>0.0561 | 578 87<br>- ↑<br>**0.0004*** | 605 136<br>- ↑<br>**<0.0001*** |
| PTX3 | NO<br>mRNA<br>P.value | 267 476<br>↑ -<br>**0.0006*** | 332 358<br>- ↓<br>**0.0264*** | 187 530<br>↑ -<br>**<0.0001*** | 243 470<br>↑ -<br>**<0.0001*** | 396 109<br>↑ -<br>**<0.0001*** | 578 87<br>- ↑<br>**<0.0001*** | 605 136<br>- ↑<br>**<0.0001*** |
| TNFAIP6 | NO<br>mRNA<br>P.value | 267 476<br>↑ -<br>**0.0001*** | 332 358<br>- -<br>0.2072 | 187 530<br>- ↑<br>**0.0083*** | 243 470<br>- ↑<br>**0.0004*** | 396 109<br>- -<br>0.1385 | 578 87<br>↑ -<br>**0.0063*** | 605 136<br>- -<br>0.0889 |
| TNFAIP8 | NO<br>mRNA<br>P.value | 267 476<br>- -<br>0.0709 | 332 358<br>- -<br>0.1561 | 187 530<br>↑ -<br>**0.0004*** | 243 470<br>↑ -<br>**0.0157*** | 396 109<br>- -<br>0.9080 | 578 87<br>- ↑<br>**0.0241*** | 605 136<br>- ↑<br>**0.0071*** |
| STEAP4 | NO<br>mRNA<br>P.value | 267 476<br>- -<br>0.0939 | 332 358<br>- ↑<br>**0.0091*** | 187 530<br>- -<br>0.1892 | 243 470<br>- ↑<br>**0.0015*** | 396 109<br>- -<br>0.0715 | 578 87<br>- -<br>0.7019 | 605 136<br>↑ -<br>**0.0424*** |

Abbreviations: BC, breast cancer; ER, estrogen receptor; HER-2, human epidermal growth factor 2; IHC, immunohistochemistry; No, number; PR, progesterone receptor; (a Welch's tests; * P < 0.05; ↑ means upregulated; ↓ means downregulated)

was a significant correlation (p < 0.0001) between the higher SBR grade and the higher mRNA levels of TNFAIP3, PTX3, TNFAIP8, and the lower mRNA levels of EFNA1, TNFAIP6, and STEAP4. As seen in Fig 2C, there was a significant correlation (p < 0.0001) between the higher SBR grade and the higher mRNA levels of TNFAIP3, PTX3, TNFAIP8, and the lower mRNA levels of EFNA1, TNFAIP6, and STEAP4. Additionally, the differential expression of TNFAIPs across molecular subtypes (Luminal A, Luminal B, HER2 +, and TNBC), as detailed in Fig 2B and Table 1, highlights subtype-specific patterns that warrant further exploration to address their clinical relevance.

### 3.3.  The prognostic value of TNFAIPs in patients with BC

According to the Kaplan–Meier curves, higher mRNA expression of TNFAIP2, TNFAIP3, and TNFAIP8 (GG2−1) was significantly linked to better OS after FDR correction (FDR < 0.05). Furthermore, elevated mRNA levels of TNFAIP1

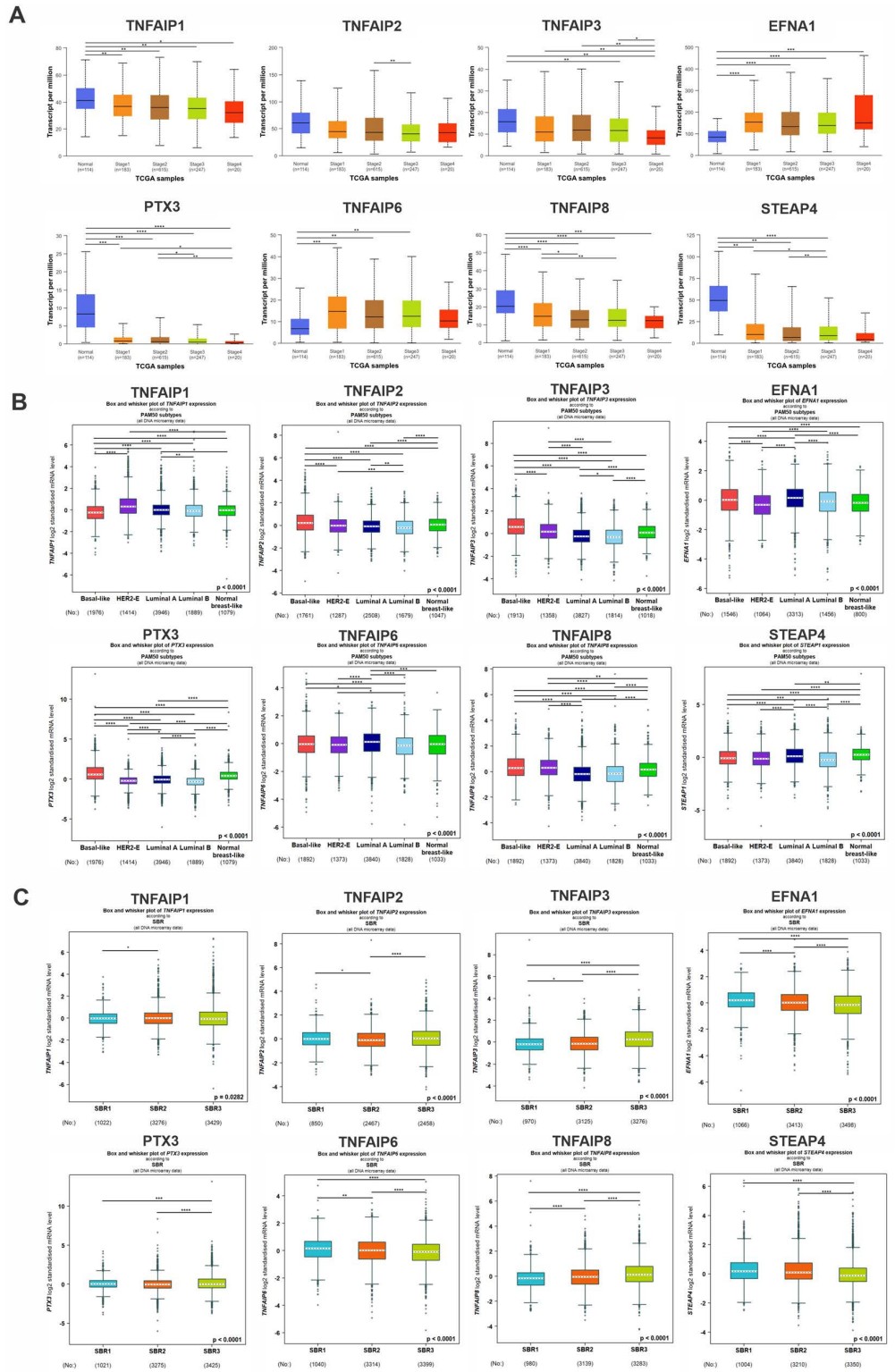

**Fig 2. A: Boxplots from the UALCAN database showing the transcriptional levels of the TNFAIP family according to BC stage.** The y-axis represents transcripts per million (TPM) expression levels, while the x-axis categorizes samples into Normal (n = 114), Stage 1 (n = 183), Stage 2 (n = 615), Stage 3 (n = 247), and Stage 4 (n = 20) groups. The colors indicate the following: blue for Normal samples, orange for Stage 1, brown for Stage 2, green

for Stage 3, and red for Stage 4. **B**: Based on the molecular subtypes found in the Bc-GenExMiner database, boxplots depicted the expression of the TNFAIP family in BC. The x-axis represents PAM50 subtypes (Basal-like, HER2-E, Luminal A, Luminal B, Normal breast-like) with sample sizes and y-axis shows Standardized mRNA expression level, with colors indicating Basal-like (red), HER2-E (purple), Luminal A (blue), Luminal B (cyan), and Normal breast-like (green). **C**: BC expression of the TNFAIP family, as reported by SBR in the Bc-GenExMiner (v5.0) database. The x-axis in each panel shows the corresponding clinical category (stage, subtype, or SBR), and the y-axis indicates relative mRNA expression levels, with colors representing SBR1 (cyan), SBR2 (orange), and SBR3 (green). (*P < 0.05, **P < 0.01, ***P < 0.001, ****P < 0.0001).

(EDP1), TNFAIP2, TNFAIP3, and STEAP4 exhibited a strong correlation with favorable RFS following FDR adjustment (FDR < 0.05) (Fig 3).

### 3.4. Genomic alterations and GO enrichment analysis of TNFAIPs in BC patients

The cBioPortal database was used for analyzing genomic changes in the TNFAIP family. According to the findings, TNFAIP family genes were altered in 348 (32%) of the 1101 BC patients (Fig 4A and 4B) displayed the alteration frequency. As a result, each of TNFAIP1, 3, and 8 was mutated in 4% of BC patients. The alteration rates of TNFAIP2, EFNA1, PTX3, TNFAIP6 and STEAP4 were 5%, 15%, 5%, 3%, and 3%, respectively. Additionally, we used Cytoscape to map and display the 50 most frequently altered neighboring genes that are co-expressed with the TNFAIPs in BC, as part of a PPI network study (Fig 4C). Moreover, we utilized GeneMANIA to identify potential interaction genes of the TNFAIP family. Fig 4D illustrates the nodes associated with the eight TNFAIP genes. Furthermore, Table 2 lists the genomic alterations of the top 10 genes that are most commonly altered with members of the TNFAIP family in BC patients.

Furthermore, the Enrichr database was used to perform enrichment analysis of TNFAIPs and their frequently altered neighbor genes. The findings of the GO analysis demonstrated that for biological processes, TNFAIPs and their neighbor genes were markedly enriched in the regulation of cell adhesion (Fig 5A). Voltage-Gated Potassium Channel Activity was observed to be the most often enriched molecular function among TNFAIPs members and their neighbor genes (Fig 5B). Furthermore, the KEGG analysis revealed that TNFAIP3 is specifically enriched in the NF-κB pathway, which plays a crucial role in regulating inflammatory and immune responses, and given TNFAIP3's regulatory role in this pathway, it may influence tumor progression. Additionally, considering the TNF-α-induced origin of TNFAIPs, the TNF signaling pathway is also suggested as a related pathway that warrants further investigation. These findings, alongside the observed enrichment in PI3K-Akt, MAPK, and apoptosis pathways, highlight a broad network of TNFAIP-driven signaling, emphasizing the multifaceted role of this family in cellular regulation (Fig 5C).

### 3.5. Prediction of transcription factor and miRNA associated with TNFAIP family

S1 and S2 Tables highlight transcription factors (TFs) and miRNAs that regulate TNFAIP family members, respectively. These data were obtained from the ChEA and miRTarBase databases via Enrichr. While several candidates showed nominal p-values < 0.05, none remained statistically significant after multiple testing correction (FDR-adj p- value > 0.05), indicating no strong evidence for regulation at the family-wide level in this dataset. Additionally, a Kaplan-Meier plotter was used in BC to evaluate the prognostic significance of the obtained TFs and miRNAs. Better overall survival was found to be significantly correlated with higher levels of the transcription factors NR3C1, NFKB1, CEBPD, and AR in BC patients. On the contrary, higher CEBPB mRNA level was linked to shorter OS in BC patients (all P < 0.05) (S1 Fig). According to Kaplan-Meier curves, there was a strong correlation between high expression of hsa-miR-23c and a shorter OS, and enhanced expression of hsa-miR-654-5p with a longer OS in BC patient. These associations remained statistically significant after Benjamini–Hochberg FDR correction (FDR-adj p- value < 0.05) (S2 Fig).

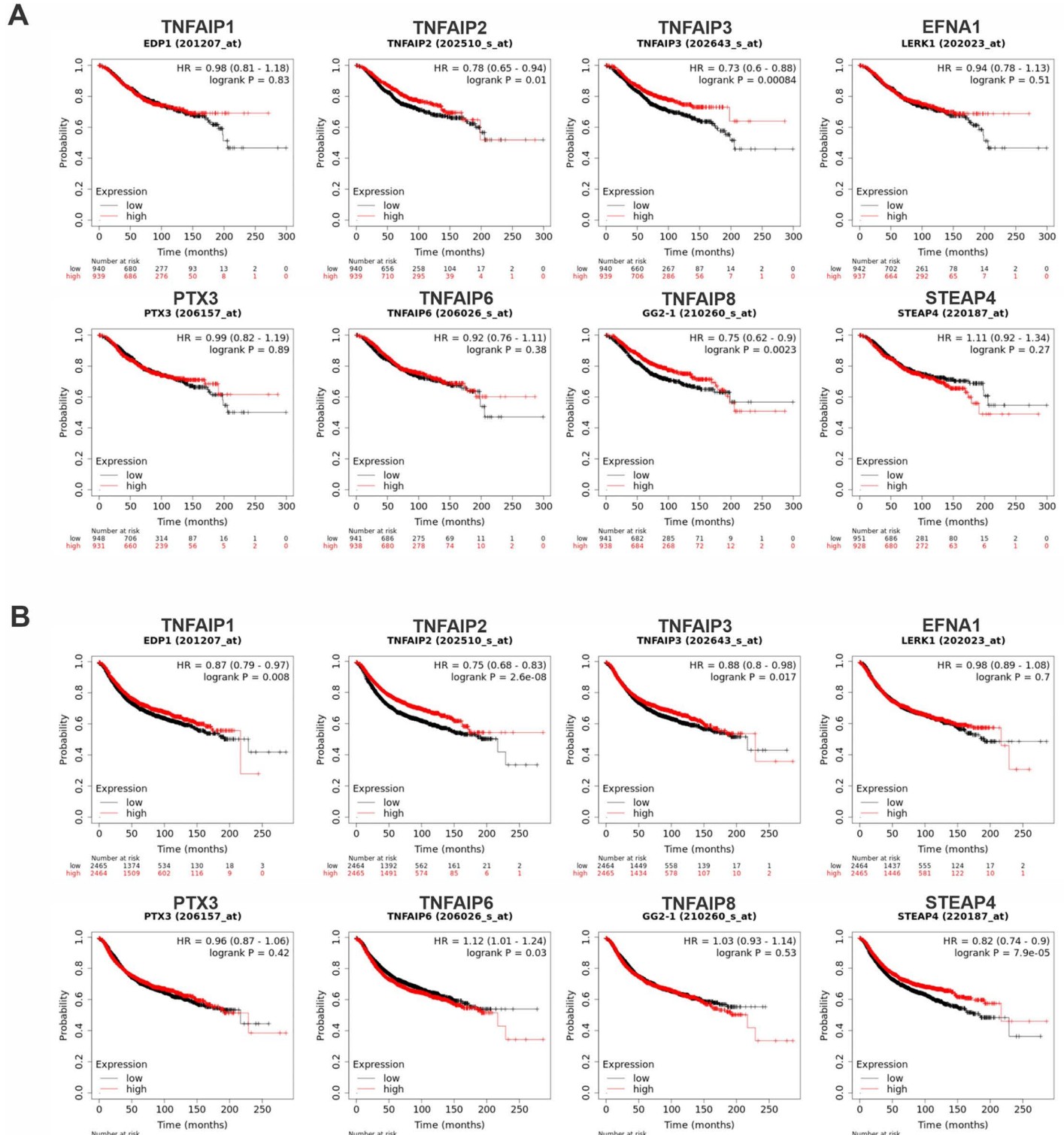

**Fig 3. The prognostic significance of TNFAIP family in BC (Kaplan-Meier plotter).** The TNFAIP family's mRNA expression is associated with OS (A) and RFS (B) in BC patients. Red and black lines represent survival curves of the patient groups with values higher and lower than the median expression levels of the target genes, respectively. X-axis: time (months); Y-axis: survival probability. (HR = hazard ratio, OS = overall survival, RFS = relapse-free survival) (TNFAIP1 (EDP1), EFNA1 (LERK1), PTX3, TNFAIP8 (GG2−1)).

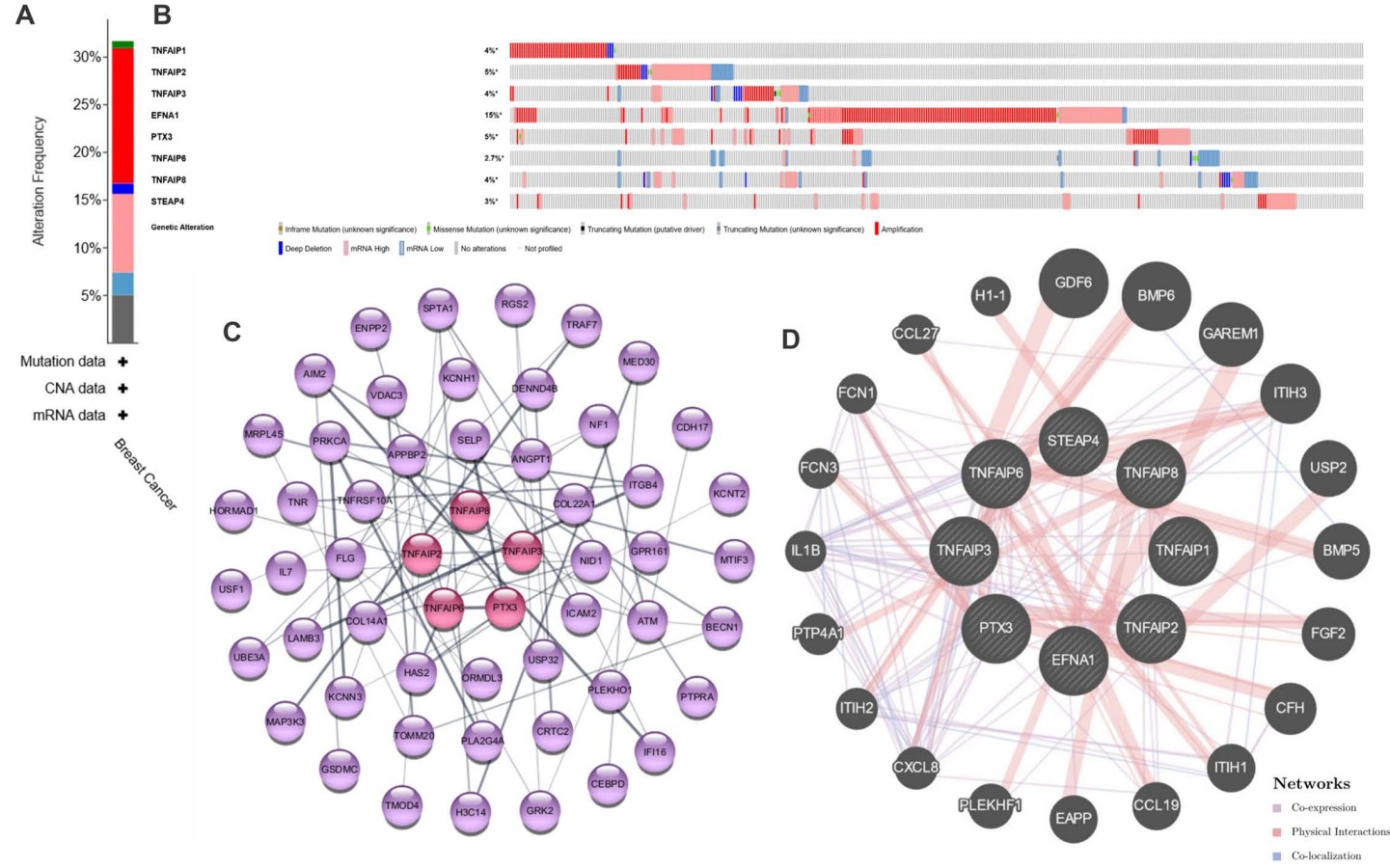

**Fig 4. A, B: These figures show TNFAIP family members genomic alterations in BC patients (cBioPortal) with colors representing Mutation (green), Amplification (red), Deep Deletion (blue), mRNA High (pink), mRNA Low (light blue), and Multiple Alterations (gray). C**: TNFAIP's protein–protein interaction (PPI) network (except TNFAIP1, EFNA1 and STEAP4 (no interaction)) (highlighted in red at the center) and the 50 most often co-expressed genes (represented in purple) in BC (STRING). The interactions are displayed as gray lines. **D**: The TNFAIP family's gene-gene interaction network in GeneMANIA. the TNFAIP family as the central genes connected by lines to other genes represented in gray, with interaction types indicated as Co-expression (purple), Physical Interactions (pink), and Co-localization (blue).

Collectively, these results suggest that while the in-silico prediction of TF/miRNA regulation did not reach family-wide significance, select TFs and miRNAs show prognostic relevance in BC, highlighting candidates for further experimental validation.

### 3.6. Single CpG methylation of TNFAIP family members and its prognostic significance in BC patients

The heatmaps representing the DNA methylation of the TNFAIP family were examined and are displayed in Fig 6. The greatest level of DNA methylation was identified in cg23245800 of TNFAIP1, cg03572388 of TNFAIP2, cg11812071 of TNFAIP3, cg12052789 of EFNA1, cg01035238 of TNFAIP6, cg01915433 of TNFAIP8 and cg09379345 of STEAP4. Furthermore, we found that the 5 CpGs of TNFAIP1, 15 CpGs of TNFAIP2, 22 CpGs of TNFAIP3, 20 CpGs of EFNA1, 6 CpGs of TNFAIP6, 32 CpGs of TNFAIP8 and 12 CpGs of STEAP4 were substantially correlated with prognosis of BC patients (S3 Table and S3 Fig).

**Table 2. The top 10 commonly changed genes in BC with the TNFAIP family.**

| Gene symbol | Total alteration (%) |
| --- | --- |
| MED30 | 25 |
| GSDMC | 22 |
| COL14A1 | 21 |
| ENPP2 | 20 |
| HAS2 | 20 |
| VDAC3 | 20 |
| ANGPT1 | 20 |
| COL22A1 | 19 |
| HORMAD1 | 18 |
| USP32 | 18 |

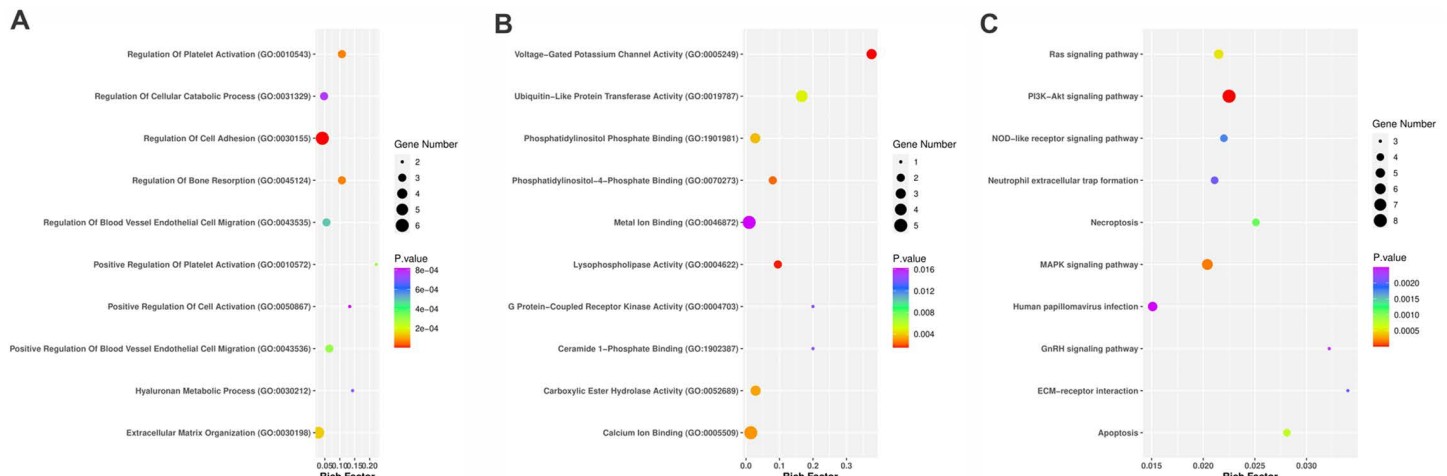

**Fig 5. The enrichment analysis of TNFAIP family and 50 top frequently altered neighbor genes in BC. Enrichment scatter plots of biological processes (A), molecular functions (B) and KEGG (C) for TNFAIP family.** The rich factor is displayed on the x-axis, while the biological processes, molecular functions and KEGG pathway terms are shown on the y-axis. X-axis: rich factor (degree of enrichment); Y-axis: GO terms or pathway names and colors indicating P-values with gene numbers marked accordingly.

### 3.7. Correlation between TNFAIPs and immune cell infiltration in BC

The involvement of TNFAIPs in immune cell infiltration and inflammatory responses influences the clinical outcome of BC patients. Consequently, we conducted a comprehensive investigation of the relationship between the expression of TNFAIPs and the infiltration of immune cells using the TIMER database. According to the findings, there was a positive correlation (p < 0.05) between the expression of TNFAIP2, TNFAIP3, TNFAIP8 and the infiltration of six different types of immune cells, including B cells, CD8 + T cells, CD4 + T cells, macrophage, neutrophil, and DCs.

Our findings demonstrated that expression of TNFAIP2, TNFAIP3, and TNFAIP8 was positively associated with the infiltration of six immune cell types (B cell, CD8 + T cell, CD4 + T cell, macrophage cells, neutrophil cells, and DCs) (p < 0.05).

In addition, the TNFAIP family showed the strongest correlation between TNFAIP3 expression levels and CD4 + T cell (Cor = 0.614, P = 1.04e − 100), neutrophils (Cor = 0.741, P = 5.15e − 166), and DCs (Cor = 0.677, P = 1.38e − 128) infiltration in BC patients (S4 Fig).

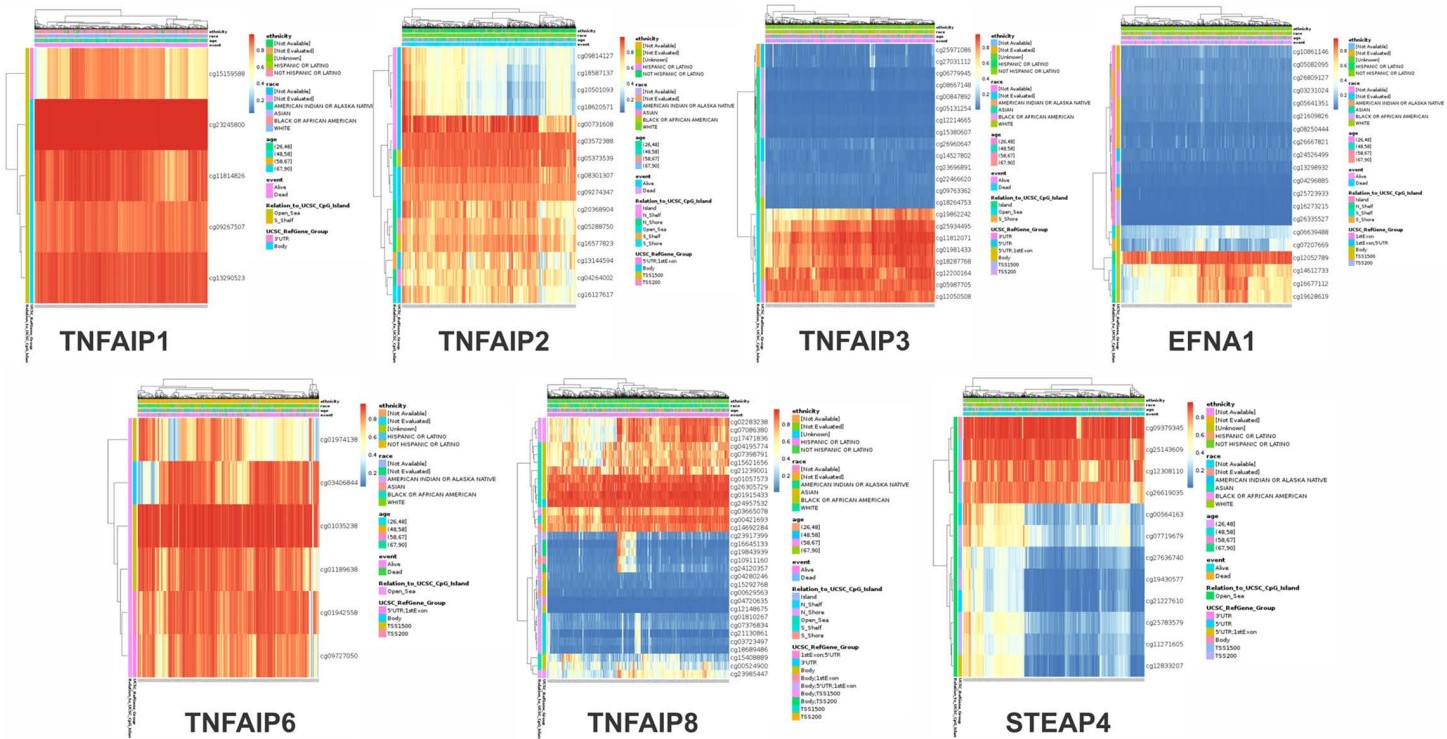

**Fig 6. The heatmap shows the TNFAIP family's CpG methylation levels (except PTX3 (no data)) in BC patients (MethSurv).** The patients are shown in columns, while the CpGs are shown in rows. The continuous variable representing methylation levels (1 = fully methylated; 0 = totally unmethylated) ranges from high expression to low expression in a color from red to blue. To depict the ethnicity, race, age, event, relationship to UCSC_CpG_island, and UCSC_refGene_Group, a variety of colorful side boxes were utilized.

### 3.8. Drug sensitivity analysis

Using the GDSC and CTRP databases, we investigated the association between the expression of the TNFAIP family and drug sensitivity, aiming to determine the response of the TNFAIP family to chemotherapy. Following GSCA and CTRP analysis, a bubble plot was utilized to display the drug sensitivity of TNFAIP1, TNFAIP2, TNFAIP3, EFNA1, PTX3, TNFAIP6, and TNFAIP8. Depending on the type of drug utilized, there was a correlation between drug resistance and both high and low expression of TNFAIP1, TNFAIP2, EFNA1, and TNFAIP6. A high expression level of PTX3 was linked to drug resistance, and low levels of TNFAIP3 and TNFAIP8 expression were also linked to drug resistance (S5A Fig). According to the CTRP database, high expression of TNFAIP1, TNFAIP2, EFNA1, PTX3 and TNFAIP6, as well as low expression of TNFAIP3 and TNFAIP8, were linked to drug resistance (S5B Fig).

A summary of the key findings related to the expression, prognostic significance, immune infiltration, and drug sensitivity of TNFAIP family members in BC is presented in Table 3.

### 3.9. Protein-level validation of TNFAIP family members in BC

To validate the transcriptomic findings, immunohistochemistry (IHC) images for TNFAIP family proteins were examined from the Human Protein Atlas (Fig 7). Consistent with our RNA-seq analysis, TNFAIP6 and EFNA1 showed notably stronger cytoplasmic and membranous staining in breast carcinoma samples compared to normal tissues, indicating upregulation at the protein level. Other family members, including TNFAIP1, TNFAIP2, TNFAIP3, PTX3, and TNFAIP8, displayed weak or comparable staining intensity between normal and malignant tissues, suggesting no major alteration in protein

**Table 3. Summary of the key findings related to TNFAIP family in BC.**

| Gene | Expression in BC vs. Normal | Prognostic value (OS/RFS) | Immune infiltration | Drug sensitivity | Key mechanism/ Subtype* |
|---|---|---|---|---|---|
| TNFAIP1 | ↓ | −/↑RFS | None | High expression→drug resistance | Apoptosis, NF-κB |
| TNFAIP2 | ↓ | ↑OS,↑RFS | ↑ B, CD4+, CD8+, MΦ, Neutrophil, DCs | High expression→drug resistance | Rac1-ERK-AP1-HIF1α in TNBC |
| TNFAIP3 | ↓ | ↑OS,↑RFS | ↑ CD4+ (0.614), Neutrophils (0.741), DCs (0.677) | Low expression→drug resistance | NF-κB suppression |
| EFNA1 | ↑ | −/− | None | High expression→drug resistance | Luminal A enrichment |
| PTX3 | ↓ | −/− | None | High expression→drug resistance | Inflammation in TNBC |
| TNFAIP6 | ↑ | −/− | None | High expression→drug resistance | Wound healing, Luminal A |
| TNFAIP8 | ↓ | ↑OS / - | ↑All six immune cell types | Low expression→drug resistance | Autophagy, metastasis |
| STEAP4 | ↓ | −/↑RFS | None | Not assessed | High SBR correlation |

Footer: *Indicates insights from referenced literature in Section 4 (Discussion). See text for specific citations.

Abbreviations: OS, Overall Survival; RFS, Relapse-Free Survival; MΦ, Macrophages; DCs, Dendritic cells;↑, Upregulated/Positive correlation;↓, Downregulated/Negative correlation; −, Not significant or not assessed

abundance. These IHC results visually corroborate our computational analysis, highlighting EFNA1 and TNFAIP6 as the key TNFAIP genes upregulated in BC. However, currently there is lacking data in HPA open-source to support STEAP4 findings.

To provide a systems-level integration of the multi-omics findings presented above, we constructed a conceptual schematic model summarizing TNFAIP family dysregulation and its associated molecular interactions in BC (Fig 8).

## 4. Discussion

### 4.1. Prognostic role of TNFAIP family in BC

Significant data support the view that the TNFAIP family members are involved in the growth, invasion, and metastasis of tumor cells. They are directly linked to the development and occurrence of many malignant cancers, such as osteosarcoma, pancreatic, gastric, colorectal, lung, and BC [8,16,34]. Building upon this background, the present study aimed to systematically evaluate the prognostic significance of the entire TNFAIP family in BC using an integrative bioinformatics framework. The integrative schematic model (Fig 8) provides a systems-level conceptual overview of how TNFAIP dysregulation may interface with NF-κB, PI3K-Akt, and TNF signaling pathways, immune infiltration patterns, and therapeutic response in BC.

Collectively, the expression-survival patterns of TNFAIP members suggest distinct biological roles within the TNFAIP network in BC. Downregulated members such as TNFAIP3 and STEAP4 appear to exert tumor-suppressive effects, likely through NF-κB and oxidative-stress regulation, whereas the upregulated EFNA1 and TNFAIP6 may promote extracellular-matrix remodeling and inflammatory processes, potentially influencing subtype-specific tumor behavior, particularly in Luminal A tumors. This mechanistic link aligns with findings that inflammatory signaling can drive ECM reorganization and create a tumor-permissive stroma [11]. Based on IHC images, the stronger cytoplasmic/membranous localization of TNFAIP6 and EFNA1 in tumor cells suggests potential roles in extracellular matrix remodeling and oncogenic

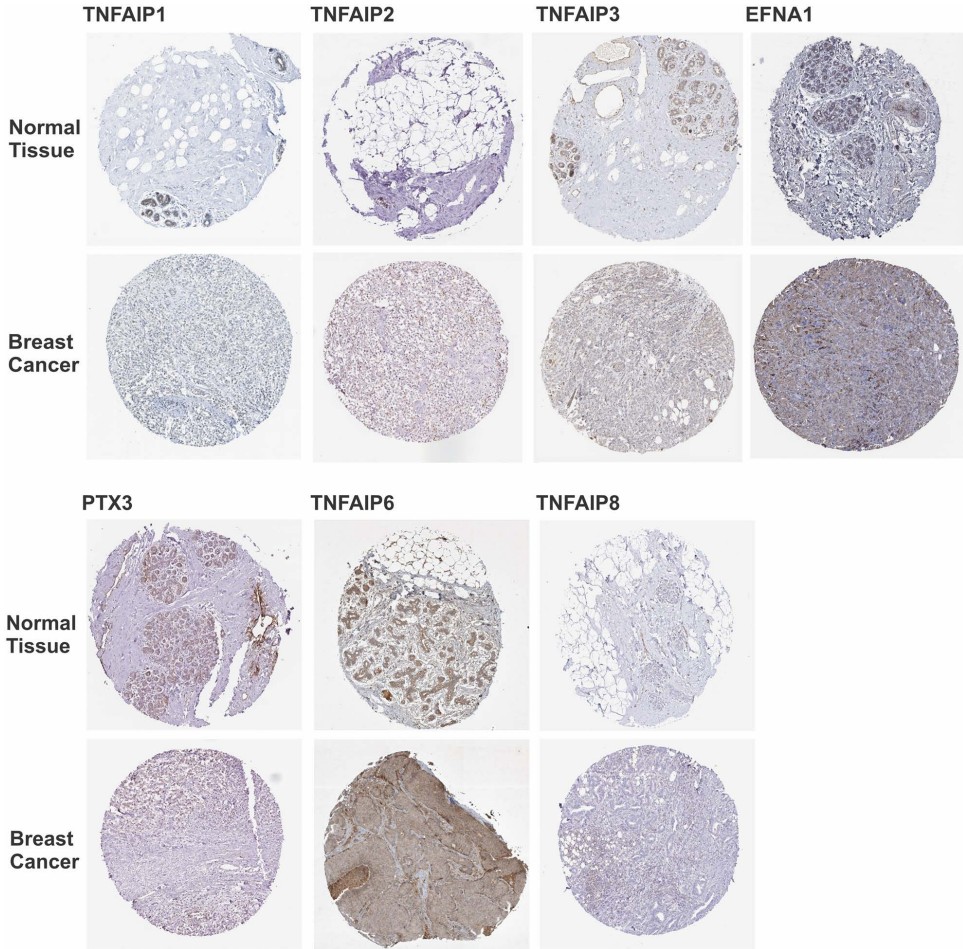

**Fig 7. Representative immunohistochemistry (IHC) images of TNFAIP family proteins in normal and breast cancer tissues retrieved from the Human Protein Atlas.** Brown DAB staining indicates protein expression level. EFNA1, and TNFAIP6 exhibit stronger staining in breast carcinoma compared with normal breast tissue, consistent with their potential upregulation during tumorigenesis. TNFAIP1, TNFAIP2, TNFAIP3, PTX3, and TNFAIP8 show weak or comparable expression between normal and cancer tissues. Images were obtained using validated antibodies from the Human Protein Atlas.

signaling, respectively. These findings emphasize the selective functional involvement of specific TNFAIP members in BC pathobiology.

Beyond the TNFAIP genes themselves, several of the top co-altered genes identified in Table 2 have established roles in cancer biology, providing additional biological context for their coordinated dysregulation. The progression of BC has been linked to hyaluronan synthase 2 (HAS2), whose overexpression is associated with increased extracellular matrix remodeling and invasive behavior. According to functional research, HAS2 promotes the invasion of BC cells and may alter the dynamics of the tumor microenvironment [35]. ENPP2, which encodes autotaxin, contributes to lysophosphatidic acid (LPA) signaling, which has been found to promote BC growth and inflammatory signaling within the tumor microenvironment [36]. Elevated expression of COL22A1 has been linked to increased migration, proliferation, and apoptosis resistance in glioblastoma and other malignancies [37], while GSDMC, a member of the gasdermin family, has been associated with inflammatory signaling, pyroptosis regulation, tumor proliferation, and stemness in multiple tumor types,

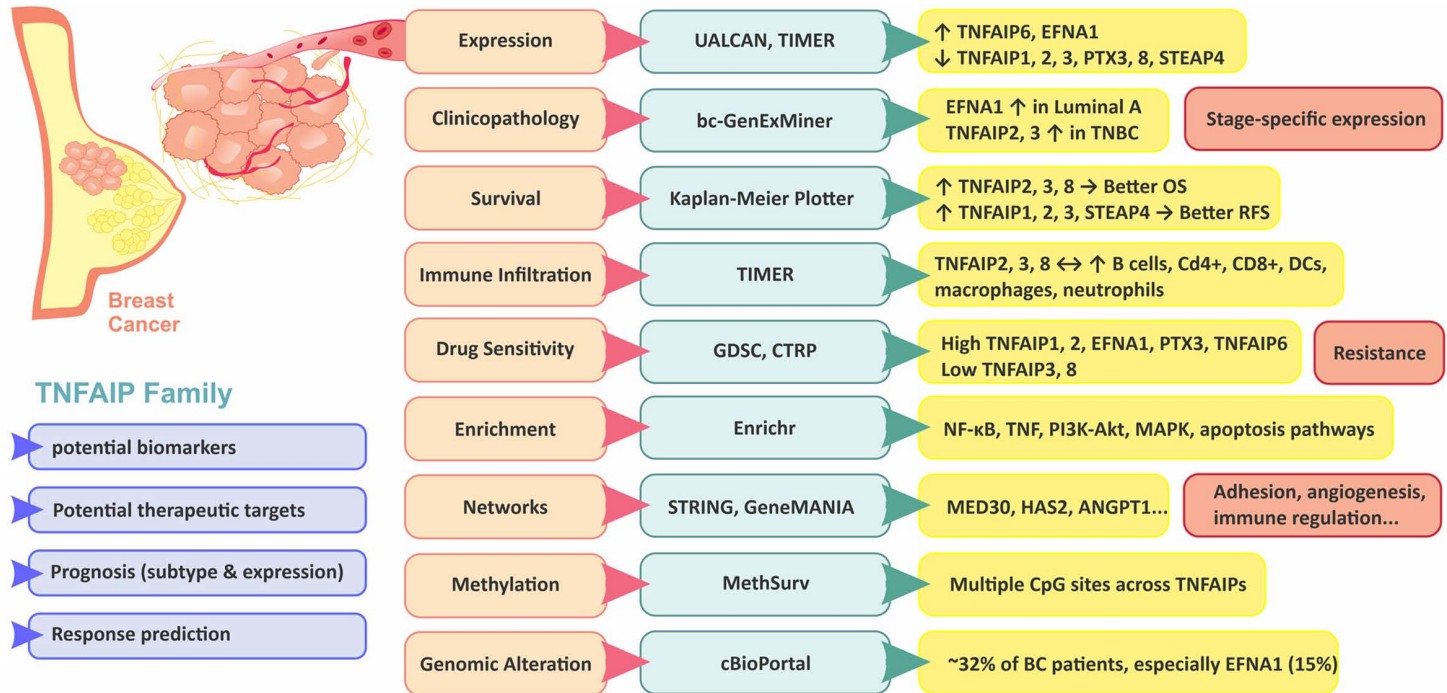

**Fig 8. Integrative conceptual model of TNFAIP family dysregulation in BC. This schematic integrates multi-omics findings from the present study, including gene expression patterns, clinicopathological associations, survival analyses, immune infiltration correlations, enrichment results, methylation profiles, genomic alterations, and drug-sensitivity associations.** Solid arrows indicate associations directly inferred from the current computational analyses across publicly available databases. Dashed arrows represent biological mechanisms supported by previously published experimental studies and incorporated to provide biological context. The proposed model is conceptual and hypothesis-generating, and does not imply direct causal relationships derived solely from the present in silico analyses.

including BC [38]. VDAC3 is a voltage-dependent anion channel implicated in mitochondrial metabolism and cell survival, processes frequently altered in cancer [39].

Similarly, ANGPT1 encodes angiopoietin-1, a regulator of angiogenesis and vascular stability, which can influence tumor progression. HORMAD1, typically restricted to germ cells, is mis-expressed in a subset of TNBCs, which correlates with genomic instability and may contribute to aggressive phenotypes [40,41]. The deubiquitinating enzyme USP32 is becoming a viable target for treatment and a tumor promoter because of its functions in oncogenic signaling networks and protein stability [42]. MED30, a Mediator complex subunit involved in transcriptional regulation, and other extracellular matrix-related collagens (e.g., COL14A1) likely contribute to regulatory networks affecting proliferation, invasion, and matrix organization [43,44].

Collectively, the coordinated alteration of these genes points toward convergent mechanisms involving signaling, matrix remodeling, metabolism, angiogenesis, and genomic instability that may functionally cooperate with TNFAIP family dysregulation in BC progression.

In our GO enrichment analysis, voltage-gated potassium channel activity was one of the most significantly enriched molecular functions among TNFAIP members and their frequently altered neighbor genes. Voltage-gated potassium channels (VGKCs), which include multiple subtypes of Kv family proteins, have been shown to be expressed and functionally active in various cancers, where they contribute to neoplastic processes such as cell proliferation, migration, cell cycle progression, and apoptotic regulation [45]. These ion channels have been proposed as part of a class of 'oncochannels' with emerging roles in tumor progression and as potential therapeutic targets in cancer research. The enrichment

of VGKC-related functions in our analysis suggests that ion-channel–mediated regulatory mechanisms may represent an additional layer through which TNFAIP-associated networks influence tumor behavior, although further functional validation is warranted.

Prior studies have shown that EFNA1 overexpression is significantly associated with poor prognosis in colorectal cancer (CRC), including higher rates of relapse and cancer-related death [46,47]. In our BC cohort, EFNA1 was consistently upregulated, supporting its potential involvement in tumor progression. While its prognostic relevance appears to vary across tumor types and molecular contexts [48,49], these observations suggest that EFNA1 may exert context-dependent biological effects across malignancies The enrichment of TNFAIPs in the PI3K-Akt pathway, as observed in our KEGG analysis (Fig 5C), indicates a potential mechanism whereby altered expression (e.g., upregulation of EFNA1 and TNFAIP6) may promote cell survival, tumor growth, and resistance to apoptosis [15]. These pathway-level findings provide a biologically plausible framework through which TNFAIP-associated signaling networks may influence tumor behavior and subtype-specific characteristics in BC. The molecular mechanisms underlying the influence of TNFAIP family members on BC prognosis are multifaceted. TNFAIP1, which exhibits favorable RFS in our study, is known to inhibit the NF-κB signaling pathway, a key regulator of inflammation and cell survival, potentially reducing tumor aggressiveness through apoptosis induction [50,51]. A 2020 study reported that TNFAIP1 expression was significantly decreased in hepatocellular carcinoma (HCC) tissues, with lower TNFAIP1 expression correlating with more advanced tumor grade, suggesting TNFAIP1 levels may also inform prognosis in liver cancer [51].

Our study suggests that both TNFAIP1 and TNFAIP2 serve as positive prognostic markers in BC outcomes through their mRNA expression level. For TNFAIP2, our findings of improved OS and RFS align with its regulation of MAPK signaling, which modulates cell death and proliferation, suggesting a protective role against BC progression [52].

Accordingly, based on a 2022 Pan-cancer analysis TNFAIP2 expression is associated with the regulation of MAPK signaling as well as cell death and has multifaceted prognostic value across cancers, with its upregulation linked to favorable OS in several cancers such as bladder carcinoma and sarcoma, though in some cancers like AML, it associates with poor prognosis [52]. Lin Jia et al. (2018) indicated that KLF5 induces TNFAIP2 and thus stimulates the proliferation of triple-negative BC cells [17]. In terms of mechanism, TNFAIP2 interacts with two GTPases: Rac1 and Cdc42. These GTPases are known to modify the actin cytoskeleton and cell shape in BC, and their activity is increased through this interaction [12]. In addition, TNFAIP2 has been reported to enhance HIF1α transcription and promote BC angiogenesis by activating the Rac1-ERK-AP1 signaling axis, further supporting its role in tumor progression [16]. These findings collectively highlight the context-dependent yet mechanistically plausible roles of TNFAIP1 and TNFAIP2 in modulating BC progression.

Given TNFAIP3's role as an NF-κB suppressor, its lower expression in aggressive subtypes suggests a loss of anti-inflammatory control that may enhance tumor progression [53]. Sharif-Askari et al. (2021) demonstrated that TNFAIP3/A20 expression is tumor-specific and varies among subtypes. Importantly, their study associated TNFAIP3 overexpression with poorer outcomes in endocrine-treated patients, suggesting that TNFAIP3's prognostic impact is context-dependent [54]. Our subtype-stratified analysis (Fig 2) similarly showed higher TNFAIP3 expression in Luminal A/B tumors and lower expression in Basal-like/TNBC subtypes, supporting its role as a subtype-specific regulatory factor in BC. Given the large number of statistical comparisons across independent databases, our findings should be interpreted in the context of potential multiple-testing effects

## 4.2. Immune associations and tumor microenvironment

The immune infiltration patterns (Table 3 and S4 Fig) of TNFAIP2, TNFAIP3, and TNFAIP8 indicate their integration within BC's immune-regulatory landscape Among these, TNFAIP3 demonstrated particularly strong correlations with dendritic-cell and T-cell infiltration, supporting its established role as a key modulator of inflammatory signaling. Mechanistically,

TNFAIP3 negatively controls NF-κB signaling, which is central to inflammatory and immune responses. In CD8＋T cells, TNFAIP3 deletion increases production of inflammatory cytokines such as IFN-γ and TNF-α and enhances antitumor immunity, including better responses to PD-1 immune checkpoint blockade in cancer models [55]. These findings provide biological plausibility for the immune associations identified in our dataset.

Accordingly, TNFAIP8 regulates immune and inflammatory responses by affecting immune cell proliferation and polarization. It promotes proliferation of CD4＋T lymphocytes and modulates their function after inflammatory conditions. Consistent with prior reports, TNFAIP8 expression positively correlates with CD8＋T cell, CD4＋T cell, macrophage, and dendritic cell infiltration in tumors. In addition to immune modulation, TNFAIP8 has been reported to influence cancer cell signaling pathways associated with autophagy, drug resistance, cell survival, proliferation, and metastasis [56,57]. In our BC analysis, TNFAIP8 was downregulated in tumor tissues, and higher TNFAIP8 expression was significantly associated with improved OS, suggesting a potential protective and context-dependent role in BC. The distinct expression and survival pattern observed here may reflect cancer-type-specific functional heterogeneity. Further experimental studies will be necessary to delineate the mechanistic basis of these context-dependent effects. Also, TNFAIP2 shows immune infiltration-related expression patterns, although detailed mechanisms in BC are less well defined compared to TNFAIP3 and TNFAIP8. Previous studies have reported high TNFAIP2 expression in cancers with immune-related functional involvement [7].

Importantly, our findings are derived from bulk transcriptomic datasets, which do not permit precise attribution of TNFAIP expression to specific cellular compartments. Therefore, it remains unclear whether the observed associations reflect expression within tumor cells, infiltrating immune populations, or stromal components. Future investigations employing single-cell and spatial transcriptomic approaches will be essential to resolve the cellular origin and spatial distribution of TNFAIP family members in the breast tumor microenvironment.

### 4.3. Therapeutic implications and drug sensitivity

The observed correlations between TNFAIP expression and drug sensitivity, such as resistance patterns associated with high TNFAIP1, TNFAIP2, EFNA1 and low TNFAIP3, TNFAIP8, together with their immune-infiltration associations (e.g., TNFAIP3 with CD4＋T cells), provide a potential framework for therapeutic stratification in BC. These findings suggest that TNFAIP-associated networks may influence responsiveness to chemotherapy and targeted agents, including therapies modulating NF-κB, PI3K-Akt, and MAPK signaling pathways, and could therefore contribute to predictive biomarker development for personalized treatment strategies [58]. TNFAIP6 and EFNA1, given their upregulation in BC and potential involvement in aggressive molecular subtypes, may represent candidates for further therapeutic investigation, particularly in Luminal A and high-risk subgroups. However, this remains speculative until validated experimentally through functional studies. It should be noted that our drug-sensitivity analysis was primarily based on cancer cell line data, and patient-level pharmacogenomic validation was not available. Therefore, these findings are preliminary and hypothesis-generating, providing a foundation for future in vitro, in vivo, or clinical studies to evaluate TNFAIP family members as predictive biomarkers for therapeutic response. Conversely, the downregulation of TNFAIP1, TNFAIP2, TNFAIP3, and STEAP4, linked to favorable prognosis, suggests that strategies to upregulate their expression (e.g., gene therapy or small molecule activators) might enhance therapeutic efficacy. While speculative, this concept is consistent with their putative tumor-suppressive functions identified in our multi-omics analyses.

With respect to PTX3, Zhou et al. (2025) reported significantly reduced PTX3 mRNA and protein expression in lung adenocarcinoma (LUAD) tissues compared with adjacent non-cancerous samples [59]. Consistent with our findings of lower PTX3 protein expression in BC tissues, their study linked reduced PTX3 levels to tumor microenvironment remodeling, diagnostic relevance, and therapeutic resistance in LUAD. However, mechanistic studies show a complex, context-dependent role of PTX3 in cancer biology, involving pathways such as EMT, macrophage polarization, autophagy

regulation, and growth factor signaling pathways [60–62]. These data collectively support the view that PTX3 may function as either a protumoral or antitumoral factor depending on tumor type and microenvironmental context.

Our results indicated that higher STEAP4 expression was associated with better relapse-free survival, whereas tumors with reduced STEAP4 levels tended to exhibit higher SBR grades, consistent with a less differentiated phenotype. Interestingly, an immunohistochemical study by Abu-Farsakh et al. (2021) reported STEAP4 expression only in malignant breast tumors and found it correlated with higher tumor grade, supporting its involvement in tumor progression [63]. This apparent discrepancy may reflect differences in molecular subtype composition and detection methods: our large-scale transcriptomic analysis integrates data from multiple BC subtypes, whereas their cohort primarily comprised high-grade tumors. Together, these data highlight STEAP4's complex, context-dependent role in BC biology.

Recent studies have examined individual TNFAIP members or limited family subsets in various cancers. Lan et al. (2021) analyzed TNFAIPs in head and neck cancer [7], while Zhang et al. (2024) focused on the TNFAIP8 family in glioma [9]. In BC specifically, Ren et al. (2024) [16] and Li et al. (2025) [10] demonstrated TNFAIP2's mechanistic roles in angiogenesis and chemoresistance, respectively. However, these studies either investigated a single member or lacked multi-omics integration or considered no subtype or immune-drug integration. In contrast, our work provides a family-wide, subtype-stratified, multi-omics analysis integrating expression, methylation, immune infiltration, and drug-sensitivity data in BC, thereby offering a broader systems-level perspective that complements and extends prior experimental findings.

A key limitation of our study is that it is based exclusively on in silico analyses of publicly available datasets. While these integrative bioinformatics approaches provide valuable insights, they cannot fully substitute for experimental validation. Nevertheless, the use of FDR correction and validation across independent databases (e.g., UALCAN, TIMER, and bc-GenExMiner) enhances the robustness and consistency of the observed associations. Thus, our findings should be considered hypothesis-generating rather than conclusive. Future studies incorporating in vitro and in vivo validation experiments will be necessary to confirm the mechanistic roles and clinical utility of TNFAIP family members in BC. To establish the clinical utility of TNFAIP family members as biomarkers or therapeutic targets in BC, future studies should include in vitro experiments (e.g., quantitative PCR, Western blotting, functional assays) and in vivo validation in patient-derived samples or animal models.

## 5. Conclusion

This comprehensive multi-omics bioinformatics analysis demonstrates significant associations between TNFAIP family members and BC prognosis, immune landscape, and drug sensitivity. By integrating expression, methylation, survival, immune infiltration, and pharmacogenomic data, our study provides a systems-level characterization of TNFAIP dysregulation in BC.

Our findings indicate that TNFAIPs may serve as potential prognostic biomarkers and putative therapeutic targets in BC. In particular, the observed relationships between TNFAIP expression patterns, tumor microenvironment features, and drug-response profiles underscore their possible relevance in patient stratification and personalized therapeutic strategies.

Nevertheless, these results are derived from in silico analyses and require experimental validation in clinical specimens and functional models. Future investigations should aim to elucidate the precise molecular mechanisms underlying TNFAIP-mediated signaling and to assess their translational applicability in targeted therapy and immunotherapy settings.

Overall, this study establishes a comprehensive analytical framework that may facilitate subsequent laboratory and translational research focused on the TNFAIP family in BC.

## Supporting information

**S1 Table. Key regulated TFs of TNFAIP family in BC.**
(DOCX)

**S2 Table. Key miRNAs regulating TNFAIPs in BC.**
(DOCX)

**S3 Table. The prognostic significance of TNFAIP family single CpG methylation in patients with BC (MethSurv).**
(DOCX)

**S1 Fig. The Kaplan-Meier plotter's prognostic value for the TFs regulating TNFAIP family members.** The correlation between OS in BC patients and the mRNA expression of NR3C1, NFKB1, CEBPD, CEBPB and AR. P < 0.05 was the threshold for significance. The confidence intervals are shown in brackets. Black indicates low expression, whereas red indicates high expression. The x-axis indicates time (in months), and the y-axis represents survival probability. The hazard ratio is HR.
(DOCX)

**S2 Fig. The Kaplan-Meier plotter's prognostic value for the miRNAs regulating TNFAIP family members.** The correlation between OS in BC patients and the miRNA expression of hsa-miR-23c and hsa-miR-654-5p. p < 0.05 was the threshold for significance. The confidence intervals are shown in brackets. Black indicates low expression, whereas red indicates high expression. The x-axis indicates time (in months), and the y-axis represents survival probability. The hazard ratio is HR.
(DOCX)

**S3 Fig. Kaplan-Meier curves to illustrate the single CpG methylation of TNFAIP family's prognostic value in BC patients.** The red curve represents patients with high methylation levels, while the blue curve represents patients with low methylation levels. The x-axis indicates time (in days), and the y-axis represents survival probability.
(DOCX)

**S4 Fig. The correlation between the expression of the TNFAIP family and the level of immune cell infiltration in BC (TIMER).** This figure presents scatter plots illustrating the correlation between mRNA expression levels (log2 TPM) of TNFAIP family genes and immune cell infiltration levels in BC patients. Each subplot corresponds to a specific TNFAIP gene and evaluates the relationship with different immune cell types. The x-axis represents the infiltration level of each immune cell type, while the y-axis represents the log2 TPM expression level of the respective gene. Blue regression lines indicate the trend of correlation. Correlation coefficients (cor) and partial correlation coefficients (partial cor) are provided, with p-values indicating statistical significance.
(DOCX)

**S5 Fig. Association between TNFAIP family and sensitivity to FDA-approved drugs (GSCALite database).** A: A bubble plot was used to illustrate the relationship between the expression of the TNFAIP family and the sensitivity of the top GDSC drugs in pan-cancer. B: A bubble plot was used to illustrate the relationship between the expression of the TNFAIP family and the sensitivity of CTRP drugs (top 30) in pan-cancer. The x-axis represents different drugs, while the y-axis lists TNFAIP family genes. The color intensity and direction indicate the correlation strength and direction: purple represents negative correlation (up to −0.4), while red represents positive correlation (up to 0.5). The size of the circles reflects the False Discovery Rate (FDR) significance levels. (GDSC: Genomics of Drug Sensitivity in Cancer, CTRP: Cancer Therapeutics Response Portal, GSCALite: Gene Set Cancer Analysis).
(DOCX)

## Author contributions

**Conceptualization:** Tahereh Barati.

**Data curation:** Tahereh Barati, Madiheh Mazaheri Moghaddam, Fatemeh Mokhles, Zohreh Mirzaei, Amir Ebrahimi, Golsa Nayeb Ghanbar Hosseini.

**Formal analysis:** Tahereh Barati, Madiheh Mazaheri Moghaddam, Fatemeh Mokhles, Zohreh Mirzaei, Amir Ebrahimi, Golsa Nayeb Ghanbar Hosseini.

**Investigation:** Tahereh Barati, Madiheh Mazaheri Moghaddam, Fatemeh Mokhles, Zohreh Mirzaei, Amir Ebrahimi, Golsa Nayeb Ghanbar Hosseini.

**Methodology:** Tahereh Barati, Madiheh Mazaheri Moghaddam, Fatemeh Mokhles.

**Supervision:** Najaf Allahyari Fard.

**Validation:** Tahereh Barati.

**Visualization:** Tahereh Barati, Fatemeh Mokhles.

**Writing – original draft:** Tahereh Barati, Madiheh Mazaheri Moghaddam, Fatemeh Mokhles.

**Writing – review & editing:** Madiheh Mazaheri Moghaddam, Fatemeh Mokhles, Najaf Allahyari Fard.

## Acknowledgments

We are extremely thankful to Mr. Hossein Hozhabri for his invaluable guidance and insightful instructions in writing this manuscript; without him, this endeavor would not have been feasible. The authors thank the National Institute of Genetic Engineering and Biotechnology, Tabriz University of Medical Sciences, Zanjan University of Medical Sciences, Iran, and Institute of Biology, Leiden University, Netherland.

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
