## [Decision Letter · Decision Letter 0]

16 Feb 2026

PONE-D-25-56610Comprehensive multi-omics analysis reveals prognostic, immune, and therapeutic signatures of TNFAIP family genes in breast cancerPLOS One

Dear Dr. Allahyari Fard,

Thank you for submitting your manuscript to PLOS ONE. After careful consideration, we feel that it has merit but does not fully meet PLOS ONE’s publication criteria as it currently stands. Therefore, we invite you to submit a revised version of the manuscript that addresses the points raised during the review process.

We look forward to receiving your revised manuscript.

Kind regards,

Zeyneb Kurt, PhD

Academic Editor

PLOS One

Journal Requirements:

2. Thank you for stating the following in the Competing Interests section: [During the preparation of this work, the authors used ChatGPT to improve readability and language skills. After using this tool, the authors carefully reviewed and edited the content as needed and took full responsibility for the content of the published article.].

Reviewers' comments:

Reviewer's Responses to Questions

**Comments to the Author**

1. Is the manuscript technically sound, and do the data support the conclusions?

Reviewer #1: Yes

Reviewer #2: Partly

2. Has the statistical analysis been performed appropriately and rigorously? 

Reviewer #1: I Don't Know

Reviewer #2: N/A

3. Have the authors made all data underlying the findings in their manuscript fully available?

Reviewer #1: Yes

Reviewer #2: Yes

4. Is the manuscript presented in an intelligible fashion and written in standard English?

Reviewer #1: Yes

Reviewer #2: No

5. Review Comments to the Author

Reviewer #1: The manuscript analyzes public available data in a comprehensive way to provide multiple potential correlation between TNFAIP and breast cancer. The results of analysis are informative. I only have a couple of suggestions.

1. It is not clear which subtypes of TNFAIP come from tumor cells, which types come from immune cells, and which types come from other cells.

2. The description and discussion of data seems not enough. For examples, Table 2 lists 10 top changed genes along with change of TNFAIP, none of them is mentioned in the manuscript. In addition, voltage-gated potassium channels are mentioned but there are no detailes of subtypes of these channels and potential roles.

Reviewer #2: The authors of the manuscript presents a comprehensive in silico multi-omics analysis of the TNFAIP gene family in breast cancer (BC). They integrated transcriptomics, genomics, methylation, immune infiltration, and drug sensitivity using multiple public databases. The study is broad, systematic, and timely, and the topic is relevant. However, while the data mining is extensive, the manuscript currently overstates novelty and biological inference, lacks methodological rigor in multiple testing control, and would benefit from clearer mechanistic framing and validation logic. The substantial revisions, can help it become a good resource paper.

1. The statement that it's a first multiomics article in BC is too big to claim and needs to moderate it.

2. The stud lacks multiple testing control in crucial analysis. FDR correction is not consistent applied. This raises a chance of false positives.

3. The drug sensitivity analysis are mostly cell line based lacking patient data.

4. Although NF-κB, PI3K-Akt, and TNF signaling are highlighted, the manuscript does not provide causal models, integrated pathway analysis etc.

5. Several informations are repeated time and again in the manuscript which make it difficult to read and snthesisize a logical story.

6. Discussion need to be rewritten in more professional way.

7. A wet lab validation experiment will be really helpful.

Overall, the manuscript need a major revision to be considered for publication.

6. PLOS authors have the option to publish the peer review history of their article (what does this mean?). If published, this will include your full peer review and any attached files.

Reviewer #1: No

Reviewer #2: No

---

## [Author Response · Author response to Decision Letter 1]

12 Apr 2026

PONE-D-25-56610

Comprehensive multi-omics analysis reveals prognostic, immune, and therapeutic signatures of TNFAIP family genes in breast cancer

Dear Editor and Reviewers,

We sincerely thank you for your careful and thorough evaluation of our manuscript entitled “Comprehensive multi-omics analysis reveals prognostic, immune, and therapeutic signatures of TNFAIP family genes in breast cancer”, we greatly appreciate your detailed line-by-line feedback and constructive recommendations. Your comments have significantly improved the clarity, precision, and overall scientific presentation of our work.

In response to reviewers, we have carefully revised the manuscript and addressed all concerns as detailed below.

Response to Reviewer 1:

1. It is not clear which subtypes of TNFAIP come from tumor cells, which types come from immune cells, and which types come from other cells.

We thank the reviewer for this important comment. The datasets used in this study are based on bulk transcriptomic data (TCGA and related resources), which include mixed signals from tumor, immune, and stromal cells. Therefore, our analysis does not allow precise determination of the cellular origin of TNFAIP expression. We have clarified this limitation in the Discussion section and noted that future single-cell or spatial transcriptomic studies will be required to define cell-type-specific expression patterns.

2. The description and discussion of data seems not enough. For examples, Table 2 lists 10 top changed genes along with change of TNFAIP, none of them is mentioned in the manuscript. In addition, voltage-gated potassium channels are mentioned but there are no detailes of subtypes of these channels and potential roles.

We thank the reviewer for this constructive suggestion. We have now expanded the Discussion section to briefly describe the most relevant co-altered genes listed in Table 2 and their known roles in cancer biology. In addition, we have provided further clarification regarding the specific voltage-gated potassium channel subtypes identified in our enrichment analysis and discussed their potential functional relevance in breast cancer progression. These additions improve the biological interpretation of our findings.

We once again sincerely thank you for your valuable time, insightful comments, and constructive feedback, which have greatly contributed to improving our manuscript, and we truly appreciate the opportunity to revise our work accordingly.

Response to Reviewer 2:

1. The statement that it's a first multiomics article in BC is too big to claim and needs to moderate it.

We thank the reviewer for this thoughtful comment. We agree that the original statement was too strong and have revised the manuscript to moderate this claim accordingly.

2. The stud lacks multiple testing control in crucial analysis. FDR correction is not consistent applied. This raises a chance of false positives.

We thank the reviewer for highlighting the importance of multiple testing correction. In response, FDR adjustment using the Benjamini–Hochberg method has now been systematically applied to survival, methylation, and enrichment analyses where applicable. For analyses in which adjusted significance did not remain after FDR correction, the results have been revised accordingly and clearly indicated in the manuscript. In addition, a dedicated FDR column has been added to Supplementary Tables S1-S3 to ensure full transparency of adjusted significance values.

3. The drug sensitivity analysis are mostly cell line based lacking patient data.

We thank the reviewer for this important observation. We acknowledge that the drug sensitivity analyses performed using GDSC and CTRP databases are primarily derived from cancer cell line data, which may not fully recapitulate the complexity of the tumor microenvironment in patient samples. This represents an inherent limitation of the currently available pharmacogenomic databases. We have added this limitation explicitly to the Discussion section. Validation of these findings using patient-derived organoids or clinical cohort data with treatment response information would be a valuable direction for future studies, and we highlight this as a priority for subsequent research.

4. Although NF-κB, PI3K-Akt, and TNF signaling are highlighted, the manuscript does not provide causal models, integrated pathway analysis etc.

We thank the reviewer for this insightful comment. To address this concern, we have now added a new integrative conceptual schematic (Figure 8 replaced & previouse Figure 8, now is considered as Graphical Abstract) summarizing the multi-omics findings of our study, including expression, survival, immune infiltration, enrichment, and drug sensitivity analyses. This model provides a systems-level framework linking TNFAIP dysregulation to key signaling pathways in breast cancer.

We have clarified in the Results section and in the figure legend that this model is conceptual and hypothesis-generating. Solid arrows represent associations identified in our analyses, while dashed arrows indicate mechanisms supported by prior experimental literature. No direct causal claims are made.

5. Several informations are repeated time and again in the manuscript which make it difficult to read and snthesisize a logical story.

We thank the reviewer for this helpful comment. The manuscript has been carefully revised to reduce repetitive statements and improve overall coherence. Redundant descriptions of TNFAIP functions and pathway mechanisms were consolidated, and the Discussion section was reorganized to ensure a clearer and more logical progression of ideas. We believe these revisions have improved the readability and narrative flow of the manuscript.

6. Discussion need to be rewritten in more professional way.

We appreciate the reviewer’s suggestion. The Discussion section has been thoroughly revised to improve clarity, scientific tone, and structural organization. We refined the language to ensure a more professional and concise presentation, reduced descriptive redundancy, and strengthened the interpretative focus of the section. We believe the revised Discussion now presents a clearer, more structured, and academically rigorous interpretation of our findings.

7. A wet lab validation experiment will be really helpful.

We thank the reviewer for this valuable suggestion. We agree that experimental validation would further strengthen the biological relevance of our findings. The primary aim of this study was to provide a comprehensive multi-omics computational analysis of the TNFAIP family in breast cancer using publicly available datasets. Therefore, our results should be considered hypothesis-generating and serve as a foundation for future in vitro and in vivo investigations. Additionally, due to current resource and infrastructure limitations, experimental validation could not be performed within the timeframe of this study. We have now further clarified this point in the Discussion section.

We sincerely thank the reviewer for the thoughtful and constructive feedback. The comments have greatly contributed to improving the quality and clarity of our manuscript, and we truly appreciate the opportunity to revise our work accordingly.

---

## [Editor Report · Decision Letter 1]

15 Apr 2026

PONE-D-25-56610R1Comprehensive multi-omics analysis reveals prognostic, immune, and therapeutic signatures of TNFAIP family genes in breast cancerPLOS One

Dear Dr. Allahyari Fard,

Thank you for submitting your manuscript to PLOS ONE. After careful consideration, we feel that it has merit but does not fully meet PLOS ONE’s publication criteria as it currently stands. Therefore, we invite you to submit a revised version of the manuscript that addresses the points raised during the review process.

We look forward to receiving your revised manuscript.

Kind regards,

Zeyneb Kurt, PhD

Academic Editor

PLOS One

Journal Requirements:

Additional Editor Comments:

The authors addressed all the comments & feedback from reviewers and/or the editor sensibly.

However a link (http://bioinfo.life.hust.edu.cn/web/GSCALite/) to one of the tools used (GSCALite) is broken, which seems to have a working link on GitHub (https://github.com/chunjie-sam-liu/GSCALite).

Can the authors please revise their manuscript and correct this issue with the broken link?

---

## [Author Response · Author response to Decision Letter 2]

18 Apr 2026

Dear Editor and Reviewers,

We sincerely thank you for your careful and thorough evaluation of our manuscript entitled “Comprehensive multi-omics analysis reveals prognostic, immune, and therapeutic signatures of TNFAIP family genes in breast cancer”, we greatly appreciate your detailed line-by-line feedback and constructive recommendations. Your comments have significantly improved the clarity, precision, and overall scientific presentation of our work.

In response to journal requirements, we tried to meet every style criterion in the revised version additional to fixing declaration based on PLOS ONE policies.

In response to reviewers, we have carefully revised the manuscript and addressed all concerns as detailed below.

Response to Reviewer 1:

1. It is not clear which subtypes of TNFAIP come from tumor cells, which types come from immune cells, and which types come from other cells.

We thank the reviewer for this important comment. The datasets used in this study are based on bulk transcriptomic data (TCGA and related resources), which include mixed signals from tumor, immune, and stromal cells. Therefore, our analysis does not allow precise determination of the cellular origin of TNFAIP expression. We have clarified this limitation in the Discussion section and noted that future single-cell or spatial transcriptomic studies will be required to define cell-type-specific expression patterns.

2. The description and discussion of data seems not enough. For examples, Table 2 lists 10 top changed genes along with change of TNFAIP, none of them is mentioned in the manuscript. In addition, voltage-gated potassium channels are mentioned but there are no detailes of subtypes of these channels and potential roles.

We thank the reviewer for this constructive suggestion. We have now expanded the Discussion section to briefly describe the most relevant co-altered genes listed in Table 2 and their known roles in cancer biology. In addition, we have provided further clarification regarding the specific voltage-gated potassium channel subtypes identified in our enrichment analysis and discussed their potential functional relevance in breast cancer progression. These additions improve the biological interpretation of our findings.

We once again sincerely thank you for your valuable time, insightful comments, and constructive feedback, which have greatly contributed to improving our manuscript, and we truly appreciate the opportunity to revise our work accordingly.

Response to Reviewer 2:

1. The statement that it's a first multiomics article in BC is too big to claim and needs to moderate it.

We thank the reviewer for this thoughtful comment. We agree that the original statement was too strong and have revised the manuscript to moderate this claim accordingly.

2. The stud lacks multiple testing control in crucial analysis. FDR correction is not consistent applied. This raises a chance of false positives.

We thank the reviewer for highlighting the importance of multiple testing correction. In response, FDR adjustment using the Benjamini–Hochberg method has now been systematically applied to survival, methylation, and enrichment analyses where applicable. For analyses in which adjusted significance did not remain after FDR correction, the results have been revised accordingly and clearly indicated in the manuscript. In addition, a dedicated FDR column has been added to Supplementary Tables S1-S3 to ensure full transparency of adjusted significance values.

3. The drug sensitivity analysis are mostly cell line based lacking patient data.

We thank the reviewer for this important observation. We acknowledge that the drug sensitivity analyses performed using GDSC and CTRP databases are primarily derived from cancer cell line data, which may not fully recapitulate the complexity of the tumor microenvironment in patient samples. This represents an inherent limitation of the currently available pharmacogenomic databases. We have added this limitation explicitly to the Discussion section. Validation of these findings using patient-derived organoids or clinical cohort data with treatment response information would be a valuable direction for future studies, and we highlight this as a priority for subsequent research.

4. Although NF-κB, PI3K-Akt, and TNF signaling are highlighted, the manuscript does not provide causal models, integrated pathway analysis etc.

We thank the reviewer for this insightful comment. To address this concern, we have now added a new integrative conceptual schematic (Figure 8 replaced & previouse Figure 8, now is considered as Graphical Abstract) summarizing the multi-omics findings of our study, including expression, survival, immune infiltration, enrichment, and drug sensitivity analyses. This model provides a systems-level framework linking TNFAIP dysregulation to key signaling pathways in breast cancer.

We have clarified in the Results section and in the figure legend that this model is conceptual and hypothesis-generating. Solid arrows represent associations identified in our analyses, while dashed arrows indicate mechanisms supported by prior experimental literature. No direct causal claims are made.

5. Several informations are repeated time and again in the manuscript which make it difficult to read and snthesisize a logical story.

We thank the reviewer for this helpful comment. The manuscript has been carefully revised to reduce repetitive statements and improve overall coherence. Redundant descriptions of TNFAIP functions and pathway mechanisms were consolidated, and the Discussion section was reorganized to ensure a clearer and more logical progression of ideas. We believe these revisions have improved the readability and narrative flow of the manuscript.

6. Discussion need to be rewritten in more professional way.

We appreciate the reviewer’s suggestion. The Discussion section has been thoroughly revised to improve clarity, scientific tone, and structural organization. We refined the language to ensure a more professional and concise presentation, reduced descriptive redundancy, and strengthened the interpretative focus of the section. We believe the revised Discussion now presents a clearer, more structured, and academically rigorous interpretation of our findings.

7. A wet lab validation experiment will be really helpful.

We thank the reviewer for this valuable suggestion. We agree that experimental validation would further strengthen the biological relevance of our findings. The primary aim of this study was to provide a comprehensive multi-omics computational analysis of the TNFAIP family in breast cancer using publicly available datasets. Therefore, our results should be considered hypothesis-generating and serve as a foundation for future in vitro and in vivo investigations. Additionally, due to current resource and infrastructure limitations, experimental validation could not be performed within the timeframe of this study. We have now further clarified this point in the Discussion section.

We sincerely thank the reviewer for the thoughtful and constructive feedback. The comments have greatly contributed to improving the quality and clarity of our manuscript, and we truly appreciate the opportunity to revise our work accordingly.

---

## [Editor Report · Decision Letter 2]

20 Apr 2026

PONE-D-25-56610R2Comprehensive multi-omics analysis reveals prognostic, immune, and therapeutic signatures of TNFAIP family genes in breast cancerPLOS One

Dear Dr. Allahyari Fard,

Thank you for submitting your manuscript to PLOS ONE. After careful consideration, we feel that it has merit but does not fully meet PLOS ONE’s publication criteria as it currently stands. Therefore, we invite you to submit a revised version of the manuscript that addresses the points raised during the review process.

**The authors has not responded to the recent feedback given for their revised manuscript. The uploaded file and "responses to reviewers" seem to be repeating the responses given for the original version of the draft. Please see the comment given for their revised manuscript below and address this accordingly:**

The authors addressed all the comments & feedback from the reviewers sensibly.

However, a link (http://bioinfo.life.hust.edu.cn/web/GSCALite/) to one of the tools used (GSCALite) is broken, which seems to have a working link on GitHub (https://github.com/chunjie-sam-liu/GSCALite).

Can the authors please revise their manuscript and correct this issue with the broken link?

We look forward to receiving your revised manuscript.

Kind regards,

Zeyneb Kurt, PhD

Academic Editor

PLOS One

Journal Requirements:

Additional Editor Comments :

The authors has not responded to the recent feedback given for their revised manuscript. The uploaded file and "responses to reviewers" seem to be repeating the responses given for the original version of the draft. Please see the comment given for their revised manuscript below and address this accordingly:

The authors addressed all the comments & feedback from the reviewers sensibly.

However, a link (http://bioinfo.life.hust.edu.cn/web/GSCALite/) to one of the tools used (GSCALite) is broken, which seems to have a working link on GitHub (https://github.com/chunjie-sam-liu/GSCALite).

Can the authors please revise their manuscript and correct this issue with the broken link?

---

## [Author Response · Author response to Decision Letter 3]

22 Apr 2026

Comment: The authors addressed all the comments & feedback from the reviewers sensibly.However, a link (http://bioinfo.life.hust.edu.cn/web/GSCALite/) to one of the tools used (GSCALite) is broken, which seems to have a working link on GitHub (https://github.com/chunjie-sam-liu/GSCALite).

Can the authors please revise their manuscript and correct this issue with the broken link?

Response: Thank you for pointing out the issue regarding the broken link in our manuscript. We have now corrected the broken link for GSCALite in our manuscript. As requested, we have replaced the previous URL with the active GitHub link (https://github.com/chunjie-sam-liu/GSCALite).

---

## [Editor Report · Decision Letter 3]

23 Apr 2026

Comprehensive multi-omics analysis reveals prognostic, immune, and therapeutic signatures of TNFAIP family genes in breast cancer

PONE-D-25-56610R3

Dear Dr. Allahyari Fard,

We’re pleased to inform you that your manuscript has been judged scientifically suitable for publication and will be formally accepted for publication once it meets all outstanding technical requirements.

Kind regards,

Zeyneb Kurt, PhD

Academic Editor

PLOS One
---

## [Editor Report · Acceptance letter]

PONE-D-25-56610R3

PLOS One

Dear Dr. Allahyari Fard,

I'm pleased to inform you that your manuscript has been deemed suitable for publication in PLOS One. Congratulations! Your manuscript is now being handed over to our production team.

Kind regards,

on behalf of

Dr. Zeyneb Kurt

Academic Editor

PLOS One